# Toward Robustness against Label Noise in Training Deep Discriminative Neural Networks

**Arash Vahdat**
D-Wave Systems Inc.
Burnaby, BC, Canada
`avahdat@dwavesys.com`

## Abstract

Collecting large training datasets, annotated with high-quality labels, is costly and time-consuming. This paper proposes a novel framework for training deep convolutional neural networks from noisy labeled datasets that can be obtained cheaply. The problem is formulated using an undirected graphical model that represents the relationship between noisy and clean labels, trained in a semi-supervised setting. In our formulation, the inference over latent clean labels is tractable and is regularized during training using auxiliary sources of information. The proposed model is applied to the image labeling problem and is shown to be effective in labeling unseen images as well as reducing label noise in training on CIFAR-10 and MS COCO datasets.

## 1 Introduction

The availability of large annotated data collections such as ImageNet [1] is one of the key reasons why deep convolutional neural networks (CNNs) have been successful in the image classification problem. However, collecting training data with such high-quality annotation is very costly and time consuming. In some applications, annotators are required to be trained before identifying classes in data, and feedback from many annotators is aggregated to reduce labeling error. On the other hand, many inexpensive approaches for collecting labeled data exist, such as data mining on social media websites, search engines, querying fewer annotators per instance, or the use of amateur annotators instead of experts. However, all these low-cost approaches have one common side effect: *label noise*.

This paper tackles the problem of training deep CNNs for the image labeling task from datapoints with noisy labels. Most previous work in this area has focused on modeling label noise for multiclass classification[1] using a directed graphical model similar to Fig. 1.a. It is typically assumed that the clean labels are hidden during training, and they are marginalized by enumerating all possible classes. These techniques cannot be extended to the multilabel classification problem, where exponentially many configurations exist for labels, and the explaining-away phenomenon makes inference over latent clean labels difficult.

We propose a conditional random field (CRF) [2] model to represent the relationship between noisy and clean labels, and we show how modern deep CNNs can gain robustness against label noise using our proposed structure. We model the clean labels as latent variables during training, and we design our structure such that the latent variables can be inferred efficiently.

The main challenge in modeling clean labels as latent is the lack of semantics on latent variables. In other words, latent variables may not semantically correspond to the clean labels when the joint probability of clean and noisy labels is parameterized such that latent clean labels can take any

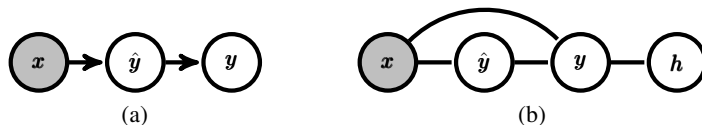

Figure 1: a) The general directed graphical model used for modeling noisy labels. $x, \hat{y}, y$ represent a data instance, its clean label, and its noisy label, respectively. b) We represent the interactions between clean and noisy labels using an undirected graphical model with hidden binary random variables ($h$).

configuration. To solve this problem, most previous work relies on either carefully initializing the conditionals [3], fine-tuning the model on the noisy set after pretraining on a clean set [4], or regularizing the transition parameters [5]. In contrast, we inject semantics to the latent variables by formulating the training problem as a semi-supervised learning problem, in which the model is trained using a large set of noisy training examples and a small set of clean training examples. To overcome the problem of inferring clean labels, we introduce a novel framework equipped with an auxiliary distribution that represents the relation between noisy and clean labels while relying on information sources different than the image content.

This paper makes the following contributions: i) A generic CRF model is proposed for training deep neural networks that is robust against label noise. The model can be applied to both multiclass and multilabel classification problems, and it can be understood as a robust loss layer, which can be plugged into any existing network. ii) We propose a novel objective function for training the deep structured model that benefits from sources of information representing the relation between clean and noisy labels. iii) We demonstrate that the model outperforms previous techniques.

## 2 Previous Work

**Learning from Noisy Labels:** Learning discriminative models from noisy-labeled data is an active area of research. A comprehensive overview of previous work in this area can be found in [6]. Previous research on modeling label noise can be grouped into two main groups: *class-conditional* and *class-and-instance-conditional* label noise models. In the former group, the label noise is assumed to be independent of the instance, and the transition probability from clean classes to the noisy classes is modeled. For example, class conditional models for binary classification problems are considered in [7, 8] whereas multiclass counterparts are targeted in [9, 5]. In the class-and-instance-conditional group, label noise is explicitly conditioned on each instance. For example, Xiao et al. [3] developed a model in which the noisy observed annotation is conditioned on binary random variables indicating if an instance's label is mistaken. Reed et al. [10] fixes noisy labels by "bootstrapping" on the labels predicted by a neural network. These techniques are all applied to either binary or multiclass classification problems in which marginalization over classes is possible. Among methods proposed for noise-robust training, Misra et al. [4] target the image multilabeling problem but model the label noise for each label independently. In contrast, our proposed CRF model represents the relation between all noisy and clean labels while the inference over latent clean labels is still tractable.

Many works have focused on semi-supervised learning using a small clean dataset combined with noisy labeled data, typically obtained from the web. Zhu et al. [11] used a pairwise similarity measure to propagate labels from labeled dataset to unlabeled one. Fergus et al. [12] proposed a graph-based label propagation, and Chen and Gupta [13] employed the weighted cross entropy loss. Recently Veit et al. [14] proposed a multi-task network containing i) a regression model that maps noisy labels and image features to clean labels ii) an image classification model that labels input. However, the model in this paper is trained using a principled objective function that regularizes the inference model using extra sources of information without the requirement for oversampling clean instances.

**Deep Structured Models:** Conditional random fields (CRFs) [2] are discriminative undirected graphical models, originally proposed for modeling sequential and structured data. Recently, they have shown state-of-the-art results in segmentation [15, 16] when combined with deep neural networks [17, 18, 19]. The main challenge in training deep CNN-CRFs is how to do inference and back-propagate gradients of the loss function through the inference. Previous approaches have focused on mean-field

approximation [16, 20], belief propagation [21, 22], unrolled inference [23, 24], and sampling [25]. The CNN-CRFs used in this work are extensions of hidden CRFs introduced in [26, 27].

# 3 Robust Discriminative Neural Network

Our goal in this paper is to train deep neural networks given a set of noisy labeled data and a small set of cleaned data. A datapoint (an image in our case) is represented by $\boldsymbol{x}$, and its noisy annotation by a binary vector $\boldsymbol{y} = \{y_1, y_2, \ldots, y_N\} \in \mathcal{Y}_N$, where $y_i \in \{0, 1\}$ indicates whether the $i^{th}$ label is present in the noisy annotation. We are interested in inferring a set of clean labels for each datapoint. The clean labels may be defined on a set different than the set of noisy labels. This is typically the case in the image annotation problem where noisy labels obtained from user tags are defined over a large set of textual tags (e.g., "cat", "kitten, "kitty", "puppy", "pup", etc.), whereas clean labels are defined on a small set of representative labels (e.g., "cat", "dog", etc.). In this paper, the clean label is represented by a stochastic binary vector $\hat{\boldsymbol{y}} = \{\hat{y}_1, \hat{y}_2, \ldots, \hat{y}_C\} \in \mathcal{Y}_C$.

We use the CRF model shown in Fig. 1.b. In our formulation, both $\hat{\boldsymbol{y}}$ and $\boldsymbol{y}$ may conditionally depend on the image $\boldsymbol{x}$. The link between $\hat{\boldsymbol{y}}$ and $\boldsymbol{y}$ captures the correlations between clean and noisy labels. These correlations help us infer latent clean labels when only the noisy labels are observed. Since noisy labels are defined over a large set of overlapping (e.g., "cat" and "pet") or co-occurring (e.g., "road" and "car") entities, $p(\boldsymbol{y}|\hat{\boldsymbol{y}}, \boldsymbol{x})$ may have a multimodal form. To keep the inference simple and still be able to model these correlations, we introduce a set of hidden binary variables represented by $\boldsymbol{h} \in \mathcal{H}$. In this case, the correlations between components of $\boldsymbol{y}$ are modeled through $\boldsymbol{h}$. These hidden variables are not connected to $\hat{\boldsymbol{y}}$ in order to keep the CRF graph bipartite.

The CRF model shown in Fig. 1.b defines the joint probability distribution of $\boldsymbol{y}, \hat{\boldsymbol{y}}$, and $\boldsymbol{h}$ conditioned on $\boldsymbol{x}$ using a parameterized energy function $E_{\boldsymbol{\theta}} : \mathcal{Y}_N \times \mathcal{Y}_C \times \mathcal{H} \times \mathcal{X} \rightarrow \mathcal{R}$. The energy function assigns a potential score $E_{\boldsymbol{\theta}}(\boldsymbol{y}, \hat{\boldsymbol{y}}, \boldsymbol{h}, \boldsymbol{x})$ to the configuration of $(\boldsymbol{y}, \hat{\boldsymbol{y}}, \boldsymbol{h}, \boldsymbol{x})$, and is parameterized by a parameter vector $\boldsymbol{\theta}$. This conditional probability distribution is defined using a Boltzmann distribution:

$$p_{\boldsymbol{\theta}}(\boldsymbol{y}, \hat{\boldsymbol{y}}, \boldsymbol{h}|\boldsymbol{x}) = \frac{1}{Z_{\boldsymbol{\theta}}(\boldsymbol{x})} \exp(-E_{\boldsymbol{\theta}}(\boldsymbol{y}, \hat{\boldsymbol{y}}, \boldsymbol{h}, \boldsymbol{x})) \tag{1}$$

where $Z_{\boldsymbol{\theta}}(\boldsymbol{x})$ is the partition function defined by $Z_{\boldsymbol{\theta}}(\boldsymbol{x}) = \sum\limits_{\boldsymbol{y} \in \mathcal{Y}_N} \sum\limits_{\hat{\boldsymbol{y}} \in \mathcal{Y}_C} \sum\limits_{\boldsymbol{h} \in \mathcal{H}} \exp(-E_{\boldsymbol{\theta}}(\boldsymbol{y}, \hat{\boldsymbol{y}}, \boldsymbol{h}, \boldsymbol{x}))$. The energy function in Fig. 1.b is defined by the quadratic function:

$$E_{\boldsymbol{\theta}}(\boldsymbol{y}, \hat{\boldsymbol{y}}, \boldsymbol{h}, \boldsymbol{x}) = -\boldsymbol{a}_{\boldsymbol{\phi}}^T(\boldsymbol{x})\hat{\boldsymbol{y}} - \boldsymbol{b}_{\boldsymbol{\phi}}^T(\boldsymbol{x})\boldsymbol{y} - \boldsymbol{c}^T\boldsymbol{h} - \hat{\boldsymbol{y}}^T\boldsymbol{W}\boldsymbol{y} - \boldsymbol{h}^T\boldsymbol{W'}\boldsymbol{y} \tag{2}$$

where the vectors $\boldsymbol{a}_{\boldsymbol{\phi}}(\boldsymbol{x}), \boldsymbol{b}_{\boldsymbol{\phi}}(\boldsymbol{x}), \boldsymbol{c}$ are the bias terms and the matrices $\boldsymbol{W}$ and $\boldsymbol{W'}$ are the pairwise interactions. In our formulation, the bias terms on the clean and noisy labels are functions of input $\boldsymbol{x}$ and are defined using a deep CNN parameterized by $\boldsymbol{\phi}$. The deep neural network together with the introduced CRF forms our *CNN-CRF* model, parameterized by $\boldsymbol{\theta} = \{\boldsymbol{\phi}, \boldsymbol{c}, \boldsymbol{W}, \boldsymbol{W'}\}$. Note that in order to regularize $\boldsymbol{W}$ and $\boldsymbol{W'}$, these matrices are not a function of $\boldsymbol{x}$.

The structure of this graph is designed such that the conditional distribution $p_{\boldsymbol{\theta}}(\hat{\boldsymbol{y}}, \boldsymbol{h}|\boldsymbol{y}, \boldsymbol{x})$ takes a simple factorial distribution that can be calculated analytically given $\boldsymbol{\theta}$ using: $p_{\boldsymbol{\theta}}(\hat{\boldsymbol{y}}, \boldsymbol{h}|\boldsymbol{y}, \boldsymbol{x}) = \prod_i p_{\boldsymbol{\theta}}(\hat{y}_i|\boldsymbol{y}, \boldsymbol{x}) \prod_j p_{\boldsymbol{\theta}}(h_j|\boldsymbol{y})$ where $p_{\boldsymbol{\theta}}(\hat{y}_i = 1|\boldsymbol{y}, \boldsymbol{x}) = \sigma(\boldsymbol{a}_{\boldsymbol{\phi}}(\boldsymbol{x})_{(i)} + \boldsymbol{W}_{(i,:)}\boldsymbol{y})$, $p_{\boldsymbol{\theta}}(h_j|\boldsymbol{y}) = \sigma(\boldsymbol{c}_{(j)} + \boldsymbol{W'}_{(j,:)}\boldsymbol{y})$, in which $\sigma(u) = \frac{1}{1+exp(-u)}$ is the logistic function, and $\boldsymbol{a}_{\boldsymbol{\phi}}(\boldsymbol{x})_{(i)}$ or $\boldsymbol{W}_{(i,:)}$ indicate the $i^{th}$ element and row in the corresponding vector or matrix respectively.

## 3.1 Semi-Supervised Learning Approach

The main challenge here is how to train the parameters of the CNN-CRF model defined in Eq. 1. To tackle this problem, we define the training problem as a semi-supervised learning problem where clean labels are observed in a small subset of a larger training set annotated with noisy labels. In this case, one can form an objective function by combining the marginal data likelihood defined on both the fully labeled clean set and noisy labeled set, and using the maximum likelihood method to learn the parameters of the model. Assume that $D_N = \{(\boldsymbol{x}^{(n)}, \boldsymbol{y}^{(n)})\}$ and $D_C = \{(\boldsymbol{x}^{(c)}, \hat{\boldsymbol{y}}^{(c)}, \boldsymbol{y}^{(c)})\}$ are two disjoint sets representing the noisy labeled and clean labeled training datasets respectively. In the

maximum likelihood method, the parameters are trained by maximizing the marginal log likelihood:

$$\max_{\boldsymbol{\theta}} \frac{1}{|D_N|} \sum_n \log p_{\boldsymbol{\theta}}(\boldsymbol{y}^{(n)}|\boldsymbol{x}^{(n)}) + \frac{1}{|D_C|} \sum_c \log p_{\boldsymbol{\theta}}(\boldsymbol{y}^{(c)}, \hat{\boldsymbol{y}}^{(c)}|\boldsymbol{x}^{(c)}) \tag{3}$$

where $p_{\boldsymbol{\theta}}(\boldsymbol{y}^{(n)}|\boldsymbol{x}^{(n)}) = \sum_{\boldsymbol{y},\boldsymbol{h}} p_{\boldsymbol{\theta}}(\boldsymbol{y}^{(n)}, \boldsymbol{y}, \boldsymbol{h}|\boldsymbol{x}^{(n)})$ and $p_{\boldsymbol{\theta}}(\boldsymbol{y}^{(c)}, \hat{\boldsymbol{y}}^{(c)}|\boldsymbol{x}^{(c)}) = \sum_{\boldsymbol{h}} p_{\boldsymbol{\theta}}(\boldsymbol{y}^{(c)}, \hat{\boldsymbol{y}}^{(c)}, \boldsymbol{h}|\boldsymbol{x}^{(c)})$. Due to the marginalization of hidden variables in log terms, the objective function cannot be analytically optimized. A common approach to optimizing the log marginals is to use the stochastic maximum likelihood method which is also known as persistent contrastive divergence (PCD) [28, 29, 25].

The stochastic maximum likelihood method, or equivalently PCD, can be fundamentally viewed as an Expectation-Maximization (EM) approach to training. The EM algorithm maximizes the variational lower bound that is formed by subtracting the Kullback–Leibler (KL) divergence between a variational approximating distribution $q$ and the true conditional distribution from the log marginal probability. For example, consider the bound for the first term in the objective function:

$$\begin{aligned}
\log p_{\boldsymbol{\theta}}(\boldsymbol{y}|\boldsymbol{x}) &\geq \log p_{\boldsymbol{\theta}}(\boldsymbol{y}|\boldsymbol{x}) - \mathrm{KL}[q(\hat{\boldsymbol{y}}, \boldsymbol{h}|\boldsymbol{y}, \boldsymbol{x})||p_{\boldsymbol{\theta}}(\hat{\boldsymbol{y}}, \boldsymbol{h}|\boldsymbol{y}, \boldsymbol{x})] &(4)\\
&= \mathbb{E}_{q(\hat{\boldsymbol{y}}, \boldsymbol{h}|\boldsymbol{y}, \boldsymbol{x})}[\log p_{\boldsymbol{\theta}}(\boldsymbol{y}, \hat{\boldsymbol{y}}, \boldsymbol{h}|\boldsymbol{x})] - \mathbb{E}_{q(\hat{\boldsymbol{y}}, \boldsymbol{h}|\boldsymbol{y}, \boldsymbol{x})}[\log q(\hat{\boldsymbol{y}}, \boldsymbol{h}|\boldsymbol{y}, \boldsymbol{x})] = \mathcal{U}_{\boldsymbol{\theta}}(\boldsymbol{x}, \boldsymbol{y}). &(5)
\end{aligned}$$

If the incremental EM approach[30] is taken for training the parameters $\boldsymbol{\theta}$, the lower bound $\mathcal{U}_{\boldsymbol{\theta}}(\boldsymbol{x}, \boldsymbol{y})$ is maximized over the noisy training set by iterating between two steps. In the Expectation step (E step), $\boldsymbol{\theta}$ is fixed and the lower bound is optimized with respect to the conditional distribution $q(\hat{\boldsymbol{y}}, \boldsymbol{h}|\boldsymbol{y}, \boldsymbol{x})$. Since this distribution is only present in the KL term in Eq. 4, the lower bound is maximized simply by setting $q(\hat{\boldsymbol{y}}, \boldsymbol{h}|\boldsymbol{y}, \boldsymbol{x})$ to the analytic $p_{\boldsymbol{\theta}}(\hat{\boldsymbol{y}}, \boldsymbol{h}|\boldsymbol{y}, \boldsymbol{x})$. In the Maximization step (M step), $q$ is fixed, and the bound is maximized with respect to the model parameters $\boldsymbol{\theta}$, which occurs only in the first expectation term in Eq. 5. This expectation can be written as $\mathbb{E}_{q(\hat{\boldsymbol{y}}, \boldsymbol{h}|\boldsymbol{y}, \boldsymbol{x})}[-E_{\boldsymbol{\theta}}(\boldsymbol{y}, \hat{\boldsymbol{y}}, \boldsymbol{h}, \boldsymbol{x})] - \log Z_{\boldsymbol{\theta}}(\boldsymbol{x})$, which is maximized by updating $\boldsymbol{\theta}$ in the direction of its gradient, computed using $-\mathbb{E}_{q(\hat{\boldsymbol{y}}, \boldsymbol{h}|\boldsymbol{x}, \boldsymbol{y})}[\frac{\partial}{\partial \boldsymbol{\theta}} E_{\boldsymbol{\theta}}(\boldsymbol{y}, \hat{\boldsymbol{y}}, \boldsymbol{h}, \boldsymbol{x})] + \mathbb{E}_{p(\boldsymbol{y}, \hat{\boldsymbol{y}}, \boldsymbol{h}|\boldsymbol{x})}[\frac{\partial}{\partial \boldsymbol{\theta}} E_{\boldsymbol{\theta}}(\boldsymbol{y}, \hat{\boldsymbol{y}}, \boldsymbol{h}, \boldsymbol{x})]$. Noting that $q(\hat{\boldsymbol{y}}, \boldsymbol{h}|\boldsymbol{y}, \boldsymbol{x})$ is set to $p_{\boldsymbol{\theta}}(\hat{\boldsymbol{y}}, \boldsymbol{h}|\boldsymbol{y}, \boldsymbol{x})$ in the E step, it becomes clear that the M step is equivalent to the parameter updates in PCD.

## 3.2 Semi-Supervised Learning Regularized by Auxiliary Distributions

The semi-supervised approach infers the latent variables using the conditional $q(\hat{\boldsymbol{y}}, \boldsymbol{h}|\boldsymbol{y}, \boldsymbol{x}) = p_{\boldsymbol{\theta}}(\hat{\boldsymbol{y}}, \boldsymbol{h}|\boldsymbol{y}, \boldsymbol{x})$. However, at the beginning of training when the model's parameters are not trained yet, sampling from the conditional distributions $p_{\boldsymbol{\theta}}(\hat{\boldsymbol{y}}, \boldsymbol{h}|\boldsymbol{y}, \boldsymbol{x})$ does not necessarily generate the clean labels accurately. The problem is more severe with the strong representation power of CNN-CRFs, as they can easily fit to poor conditional distributions that occur at the beginning of training. That is why the impact of the noisy set on training must be reduced by oversampling clean instances [14, 3].

In contrast, there may exist auxiliary sources of information that can be used to extract the relationship between noisy and clean labels. For example, non-image-related sources may be formed from semantic relatedness of labels [31]. We assume that, in using such sources, we can form an auxiliary distribution $p_{aux}(\boldsymbol{y}, \hat{\boldsymbol{y}}, \boldsymbol{h})$ representing the joint probability of noisy and clean labels and some hidden binary states. Here, we propose a framework to use this distribution to train parameters in the semi-supervised setting by guiding the variational distribution to infer the clean labels more accurately. To do so, we add a new regularization term in the lower bound that penalizes the variational distribution for being different from the conditional distribution resulting from the auxiliary distribution as follows:

$$\log p_{\boldsymbol{\theta}}(\boldsymbol{y}|\boldsymbol{x}) \geq \mathcal{U}_{\boldsymbol{\theta}}^{aux}(\boldsymbol{x}, \boldsymbol{y}) = \log p_{\boldsymbol{\theta}}(\boldsymbol{y}|\boldsymbol{x}) - \mathrm{KL}[q(\hat{\boldsymbol{y}}, \boldsymbol{h}|\boldsymbol{y}, \boldsymbol{x})||p_{\boldsymbol{\theta}}(\hat{\boldsymbol{y}}, \boldsymbol{h}|\boldsymbol{y}, \boldsymbol{x})] - \alpha \mathrm{KL}[q(\hat{\boldsymbol{y}}, \boldsymbol{h}|\boldsymbol{y}, \boldsymbol{x})||p_{aux}(\hat{\boldsymbol{y}}, \boldsymbol{h}|\boldsymbol{y})]$$

where $\alpha$ is a non-negative scalar hyper-parameter that controls the impact of the added KL term. Setting $\alpha = 0$ recovers the original variational lower bound defined in Eq. 4 whereas $\alpha \to \infty$ forces the variational distribution $q$ to ignore the $p_{\boldsymbol{\theta}}(\hat{\boldsymbol{y}}, \boldsymbol{h}|\boldsymbol{y}, \boldsymbol{x})$ term. A value between these two extremes makes the inference distribution intermediate between $p_{\boldsymbol{\theta}}(\hat{\boldsymbol{y}}, \boldsymbol{h}|\boldsymbol{y}, \boldsymbol{x})$ and $p_{aux}(\hat{\boldsymbol{y}}, \boldsymbol{h}|\boldsymbol{y})$. Note that this new lower bound is actually looser than the original bound. This may be undesired if we were actually interested in predicting noisy labels. However, our goal is to predict clean labels, and the proposed framework benefits from the regularization that is imposed on the variational distribution. Similar ideas have been explored in the posterior regularization approach [32].

Similarly, we also define a new lower bound on the second log marginal in Eq. 3 by:

$$\log p_{\boldsymbol{\theta}}(\boldsymbol{y}, \hat{\boldsymbol{y}}|\boldsymbol{x}) \geq \mathcal{L}_{\boldsymbol{\theta}}^{aux}(\boldsymbol{x}, \boldsymbol{y}, \hat{\boldsymbol{y}}) = \log p_{\boldsymbol{\theta}}(\boldsymbol{y}, \hat{\boldsymbol{y}}|\boldsymbol{x}) - \mathrm{KL}[q(\boldsymbol{h}|\boldsymbol{y})||p_{\boldsymbol{\theta}}(\boldsymbol{h}|\boldsymbol{y})] - \alpha \mathrm{KL}[q(\boldsymbol{h}|\boldsymbol{y})||p_{aux}(\boldsymbol{h}|\boldsymbol{y})].$$

**Auxiliary Distribution:** In this paper, the auxiliary joint distribution $p_{aux}(\boldsymbol{y}, \hat{\boldsymbol{y}}, \boldsymbol{h})$ is modeled by an undirected graphical model in a special form of a restricted Boltzmann machine (RBM), and is trained on the clean training set. The structure of the RBM is similar to the CRF model shown in Fig. 1.b with the fundamental difference that parameters of the model do not depend on $\boldsymbol{x}$:

$$p_{aux}(\boldsymbol{y}, \hat{\boldsymbol{y}}, \boldsymbol{h}) = \frac{1}{Z_{aux}} \exp(-E_{aux}(\boldsymbol{y}, \hat{\boldsymbol{y}}, \boldsymbol{h})) \tag{6}$$

where the energy function is defined by the quadratic function:

$$E_{aux}(\boldsymbol{y}, \hat{\boldsymbol{y}}, \boldsymbol{h}) = -\boldsymbol{a}_{aux}^T \hat{\boldsymbol{y}} - \boldsymbol{b}_{aux}^T \boldsymbol{y} - \boldsymbol{c}_{aux}^T \boldsymbol{h} - \hat{\boldsymbol{y}}^T \boldsymbol{W}_{aux} \boldsymbol{y} - \boldsymbol{h}^T \boldsymbol{W'}_{aux} \boldsymbol{y} \tag{7}$$

and $Z_{aux}$ is the partition function, defined similarly to the CRF's partition function. The number of hidden variables is set to 200 and the parameters of this generative model are trained using the PCD algorithm [28], and are fixed while the CNN-CRF model is being trained.

### 3.3 Training Robust CNN-CRF

In training, we seek $\boldsymbol{\theta}$ that maximizes the proposed lower bounds on the noisy and clean training sets:

$$\max_{\boldsymbol{\theta}} \frac{1}{|D_N|} \sum_n \mathcal{U}_{\boldsymbol{\theta}}^{aux}(\boldsymbol{x}^{(n)}, \boldsymbol{y}^{(n)}) + \frac{1}{|D_C|} \sum_c \mathcal{L}_{\boldsymbol{\theta}}^{aux}(\boldsymbol{x}^{(c)}, \boldsymbol{y}^{(c)}, \hat{\boldsymbol{y}}^{(c)}). \tag{8}$$

The optimization problem is solved in a two-step iterative procedure as follows:

**E step:** The objective function is optimized with respect to $q(\hat{\boldsymbol{y}}, \boldsymbol{h}|\boldsymbol{y}, \boldsymbol{x})$ for a fixed $\boldsymbol{\theta}$. For $\mathcal{U}_{\boldsymbol{\theta}}^{aux}(\boldsymbol{x}, \boldsymbol{y})$, this is done by solving the following problem:

$$\min_q \text{KL}[q(\hat{\boldsymbol{y}}, \boldsymbol{h}|\boldsymbol{y}, \boldsymbol{x})||p_{\boldsymbol{\theta}}(\hat{\boldsymbol{y}}, \boldsymbol{h}|\boldsymbol{y}, \boldsymbol{x})] + \alpha \text{KL}[q(\hat{\boldsymbol{y}}, \boldsymbol{h}|\boldsymbol{y}, \boldsymbol{x})||p_{aux}(\hat{\boldsymbol{y}}, \boldsymbol{h}|\boldsymbol{y})]. \tag{9}$$

The weighted average of KL terms above is minimized with respect to $q$ when:

$$q(\hat{\boldsymbol{y}}, \boldsymbol{h}|\boldsymbol{y}, \boldsymbol{x}) \propto [p_{\boldsymbol{\theta}}(\hat{\boldsymbol{y}}, \boldsymbol{h}|\boldsymbol{y}, \boldsymbol{x}) \cdot p_{aux}^{\alpha}(\hat{\boldsymbol{y}}, \boldsymbol{h}|\boldsymbol{y})]^{\left(\frac{1}{\alpha+1}\right)}, \tag{10}$$

which is a weighted geometric mean of the true conditional distribution and auxiliary distribution. Given the factorial structure of these distributions, $q(\hat{\boldsymbol{y}}, \boldsymbol{h}|\boldsymbol{y}, \boldsymbol{x})$ is also a factorial distribution:

$$q(\hat{y}_i = 1|\boldsymbol{y}, \boldsymbol{x}) = \sigma\left(\frac{1}{\alpha+1}(\boldsymbol{a}_{\boldsymbol{\phi}}(\boldsymbol{x})_{(i)} + \boldsymbol{W}_{(i,:)}\boldsymbol{y} + \alpha \boldsymbol{a}_{aux(i)} + \alpha \boldsymbol{W}_{aux(i,:)}\boldsymbol{y})\right)$$

$$q(h_j = 1|\boldsymbol{y}) = \sigma\left(\frac{1}{\alpha+1}(\boldsymbol{c}_{(j)} + \boldsymbol{W'}_{(j,:)}\boldsymbol{y} + \alpha \boldsymbol{c}_{aux(j)} + \alpha \boldsymbol{W'}_{aux(j,:)}\boldsymbol{y})\right).$$

Optimizing $\mathcal{L}_{\boldsymbol{\theta}}^{aux}(\boldsymbol{x}, \boldsymbol{y}, \hat{\boldsymbol{y}})$ w.r.t $q(\boldsymbol{h}|\boldsymbol{y})$ gives a similar factorial result:

$$q(\boldsymbol{h}|\boldsymbol{y}) \propto [p_{\boldsymbol{\theta}}(\boldsymbol{h}|\boldsymbol{y}) \cdot p_{aux}^{\alpha}(\boldsymbol{h}|\boldsymbol{y})]^{\left(\frac{1}{\alpha+1}\right)}. \tag{11}$$

**M step:** Holding $q$ fixed, the objective function is optimized with respect to $\boldsymbol{\theta}$. This is achieved by updating $\boldsymbol{\theta}$ in the direction of the gradient of $\mathbb{E}_{q(\hat{\boldsymbol{y}}, \boldsymbol{h}|\boldsymbol{x}, \boldsymbol{y})}[\log p_{\boldsymbol{\theta}}(\boldsymbol{y}, \hat{\boldsymbol{y}}, \boldsymbol{h}|\boldsymbol{x})]$, which is:

$$\frac{\partial}{\partial \boldsymbol{\theta}} \mathcal{U}_{\boldsymbol{\theta}}^{aux}(\boldsymbol{x}, \boldsymbol{y}) = \frac{\partial}{\partial \boldsymbol{\theta}} \mathbb{E}_{q(\hat{\boldsymbol{y}}, \boldsymbol{h}|\boldsymbol{x}, \boldsymbol{y})}[\log p_{\boldsymbol{\theta}}(\boldsymbol{y}, \hat{\boldsymbol{y}}, \boldsymbol{h}|\boldsymbol{x})]$$

$$= -\mathbb{E}_{q(\hat{\boldsymbol{y}}, \boldsymbol{h}|\boldsymbol{x}, \boldsymbol{y})}[\frac{\partial}{\partial \boldsymbol{\theta}} E_{\boldsymbol{\theta}}(\boldsymbol{y}, \hat{\boldsymbol{y}}, \boldsymbol{h}, \boldsymbol{x})] + \mathbb{E}_{p(\boldsymbol{y}, \hat{\boldsymbol{y}}, \boldsymbol{h}|\boldsymbol{x})}[\frac{\partial}{\partial \boldsymbol{\theta}} E_{\boldsymbol{\theta}}(\boldsymbol{y}, \hat{\boldsymbol{y}}, \boldsymbol{h}, \boldsymbol{x})], \tag{12}$$

where the first expectation (the *positive phase*) is defined under the variational distribution $q$ and the second expectation (the *negative phase*) is defined under the CRF model $p(\boldsymbol{y}, \hat{\boldsymbol{y}}, \boldsymbol{h}|\boldsymbol{x})$. With the factorial form of $q$, the first expectation is analytically tractable. The second expectation is estimated by PCD [28, 29, 25]. This approach requires maintaining a set of particles for each training instance that are used for seeding the Markov chains at each iteration of training.

The gradient of the lower bound on the clean set is defined similarly:

$$\frac{\partial}{\partial \boldsymbol{\theta}} \mathcal{L}_{\boldsymbol{\theta}}^{aux}(\boldsymbol{x}, \boldsymbol{y}, \hat{\boldsymbol{y}}) = \frac{\partial}{\partial \boldsymbol{\theta}} \mathbb{E}_{q(\boldsymbol{h}|\boldsymbol{y})}[\log p_{\boldsymbol{\theta}}(\boldsymbol{y}, \hat{\boldsymbol{y}}, \boldsymbol{h}|\boldsymbol{x})]$$

$$= -\mathbb{E}_{q(\boldsymbol{h}|\boldsymbol{y})}[\frac{\partial}{\partial \boldsymbol{\theta}} E_{\boldsymbol{\theta}}(\boldsymbol{y}, \hat{\boldsymbol{y}}, \boldsymbol{h}, \boldsymbol{x})] + \mathbb{E}_{p(\boldsymbol{y}, \hat{\boldsymbol{y}}, \boldsymbol{h}|\boldsymbol{x})}[\frac{\partial}{\partial \boldsymbol{\theta}} E_{\boldsymbol{\theta}}(\boldsymbol{y}, \hat{\boldsymbol{y}}, \boldsymbol{h}, \boldsymbol{x})] \tag{13}$$

with the minor difference that in the positive phase the clean label $\hat{\boldsymbol{y}}$ is given for each instance and the variational distribution is defined over only the hidden variables.

**Scheduling $\boldsymbol{\alpha}$:** Instead of setting $\alpha$ to a fixed value during training, it is set to a very large value at the beginning of training and is slowly decreased to smaller values. The rationale behind this is that at the beginning of training, when $p_{\boldsymbol{\theta}}(\hat{\boldsymbol{y}}, \boldsymbol{h}|\boldsymbol{y}, \boldsymbol{x})$ cannot predict the clean labels accurately, it is intuitive to rely more on pretrained $p_{aux}(\hat{\boldsymbol{y}}, \boldsymbol{h}|\boldsymbol{y})$ when inferring the latent variables. As training proceeds we shift the variational distribution $q$ more toward the true conditional distribution.

Algorithm 1 summarizes the learning procedure proposed for training our CRF-CNN. The training is done end-to-end for both CNN and CRF parameters together. In the test time, samples generated by Gibbs sampling from $p_{\boldsymbol{\theta}}(\boldsymbol{y}, \hat{\boldsymbol{y}}, \boldsymbol{h}|\boldsymbol{x})$ for the test image $\boldsymbol{x}$ are used to compute the marginal $p_{\boldsymbol{\theta}}(\hat{\boldsymbol{y}}|\boldsymbol{x})$.

---

**Algorithm 1:** Train robust CNN-CRF with simple gradient descent

---

**Input** : Noisy dataset $D_N$ and clean dataset $D_C$, auxiliary distribution $p_{aux}(\boldsymbol{y}, \hat{\boldsymbol{y}}, \boldsymbol{h})$, a learning rate parameter $\varepsilon$ and a schedule for $\alpha$
**Output:** Model parameters: $\boldsymbol{\theta} = \{\boldsymbol{\phi}, \boldsymbol{c}, \boldsymbol{W}, \boldsymbol{W'}\}$
Initialize model parameters
**while** *Stopping criteria is not met* **do**
    **foreach** *minibatch* $\{(\boldsymbol{x}^{(n)}, \boldsymbol{y}^{(n)}), (\boldsymbol{x}^{(c)}, \hat{\boldsymbol{y}}^{(c)}, \boldsymbol{y}^{(c)})\} = \texttt{getMinibatch}(D_N, D_C)$ **do**
        Compute $q(\hat{\boldsymbol{y}}, \boldsymbol{h}|\boldsymbol{y}^{(n)}, \boldsymbol{x}^{(n)})$ by Eq.10 for each noisy instance
        Compute $q(\boldsymbol{h}|\boldsymbol{y}^{(c)})$ by Eq. 11 for each clean instance
        Do Gibbs sweeps to sample from the current $p_{\boldsymbol{\theta}}(\boldsymbol{y}, \hat{\boldsymbol{y}}, \boldsymbol{h}|\boldsymbol{x}^{(\cdot)})$ for each clean/noisy instance
        $(m_n, m_c) \leftarrow$ (# noisy instances in minibatch, # clean instances in minibatch)
        $\boldsymbol{\theta} \leftarrow \boldsymbol{\theta} + \varepsilon\big(\frac{1}{m_n}\sum_n \frac{\partial}{\partial\boldsymbol{\theta}}\mathcal{U}_{\boldsymbol{\theta}}^{aux}(\boldsymbol{x}^{(n)}, \boldsymbol{y}^{(n)})\big) + \frac{1}{m_c}\sum_c \frac{\partial}{\partial\boldsymbol{\theta}}\mathcal{L}_{\boldsymbol{\theta}}^{aux}(\boldsymbol{x}^{(c)}, \boldsymbol{y}^{(c)}, \hat{\boldsymbol{y}}^{(c)})$ by Eq.12 and 13
    **end**
**end**

---

# 4 Experiments

In this section, we examine the proposed robust CNN-CRF model for the image labeling problem.

## 4.1 Microsoft COCO Dataset

The Microsoft COCO 2014 dataset is one of the largest publicly available datasets that contains both noisy and clean object labels. Created from challenging Flickr images, it is annotated with 80 object categories as well as captions describing the images. Following [4], we use the 1000 most common words in the captions as the set of noisy labels. We form a binary vector of this length for each image representing the words present in the caption. We use 73 object categories as the set of clean labels, and form binary vectors indicating whether the object categories are present in the image. We follow the same 87K/20K/20K train/validation/test split as [4], and use mean average precision (mAP) measure over these 73 object categories as the performance assessment. Finally, we use 20% of the training data as the clean labeled training set ($D_C$). The rest of data was used as the noisy training set ($D_N$), in which clean labels were ignored in training.

**Network Architectures:** We use the implementation of ResNet-50 [33] and VGG-16 [34] in Tensor-Flow as the neural networks that compute the bias coefficients in the energy function of our CRF (Eq. 2). These two networks are applied in a fully convolutional setting to each image. Their features in the final layer are pooled in the spatial domain using an average pooling operation, and these are passed through a fully connected linear layer to generate the bias terms. VGG-16 is used intentionally in order to compare our method directly with [4] that uses the same network. ResNet-50 experiments enable us to examine how our model works with other modern architectures. Misra et al. [4] have reported results when the images were upsampled to 565 pixels. Using upsampled images improves the performance significantly, but they make cross validation significantly slower. Here, we report our results for image sizes of both 224 (small) and 565 pixels (large).

**Parameters Update:** The parameters of all the networks were initialized from ImageNet-trained models that are provided in TensorFlow. The other terms in the energy function of our CRF were all

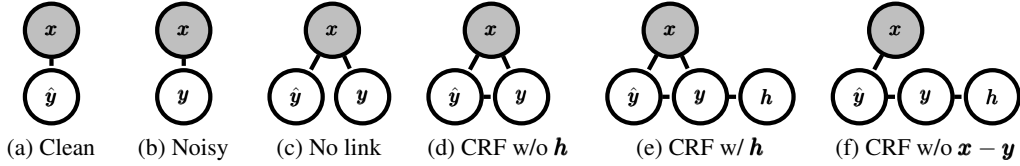

Figure 2: Visualization of different variations of the model examined in the experiments.

initialized to zero. Our gradient estimates can be high variance as they are based on a Monte Carlo estimate. For training, we use Adam [35] updates that are shown to be robust against noisy gradients. The learning rate and epsilon for the optimizer are set to (0.001, 1) and (0.0003, 0.1) respectively in VGG-16 and ResNet-50. We anneal $\alpha$ from 40 to 5 in 11 epochs.

**Sampling Overhead:** Fifty Markov chains per datapoint are maintained for PCD. In each iteration of the training, the chains are retrieved for the instances in the current minibatch, and 100 iterations of Gibbs sampling are applied for negative phase samples. After parameter updates, the final state of chains is stored in memory for the next epoch. Note that we are only required to store the state of the chains for either $(\hat{y}, h)$ or $y$. In this experiment, since the size of $h$ is 200, the former case is more memory efficient. Storing persistent chains in this dataset requires only about 1 GB of memory. In ResNet-50, sampling increases the training time only by 16% and 8% for small and large images respectively. The overhead is 9% and 5% for small and large images in VGG-16.

**Baselines:** Our proposed method is compared against several baselines visualized in Fig. 2:

- **Cross entropy loss with clean labels:** The networks are trained using cross entropy loss with the all clean labels. This defines a performance upper bound for each network.
- **Cross entropy loss with noisy labels:** The model is trained using only noisy labels. Then, predictions on the noisy labels are mapped to clean labels using the manual mapping in [4].
- **No pairwise terms:** All the pairwise terms are removed and the model is trained using analytic gradients without any sampling using our proposed objective function in Eq. 8.
- **CRF without hidden:** $W$ is trained but $W'$ is omitted from the model.
- **CRF with hidden:** Both $W$ and $W'$ are present in the model.
- **CRF without $x - y$ link:** Same as the previous model but $b$ is not a function of $x$.
- **CRF without $x - y$ link ($\alpha = 0$):** Same as the previous model but trained with $\alpha = 0$.

The experimental results are reported in Table 1 under "Caption Labels." A performance increase is observed after adding each component to the model. However, removing the $x - y$ link generally improves the performance significantly. This may be because removing this link forces the model to rely on $\hat{y}$ and its correlations with $y$ for predicting $y$ on the noisy labeled set. This can translate to better recognition of clean labels. Last but not least, the CRF model with no $x - y$ connection trained using $\alpha = 0$ performed very poorly on this dataset. This demonstrates the importance of the introduced regularization in training.

### 4.2 Microsoft COCO Dataset with Flickr Tags

The images in the COCO dataset were originally gathered and annotated from the Flickr website. This means that these image have actual noisy Flickr tags. To examine the performance of our model on actual noisy labels, we collected these tags for the COCO images using Flickr's public API. Similar to the previous section, we used the 1024 most common tags as the set of noisy labels. We observed that these tags have significantly more noise compared to the noisy labels in the previous section; therefore, it is more challenging to predict clean labels from them using the auxiliary distribution. In this section, we only examine the ResNet-50 architecture for both small and large image sizes. The different baselines introduced in the previous section are compared against each other in Table 1 under "Flickr Tags."

**Auxiliary Distribution vs. Variational Distribution:** As the auxiliary distribution $p_{aux}$ is fixed, and the variational distribution $q$ is updated using Eq. 10 in each iteration, a natural question is how

Table 1: The performance of different baselines on the COCO dataset in terms of mAP (%).

| | Caption Labels (Sec. 4.1) | | | | Flickr Tags (Sec. 4.2) | |
| | ResNet-50 | | VGG-16 | | ResNet-50 | |
| Baseline | Small | Large | Small | Large | Small | Large |
|---|---|---|---|---|---|---|
| Cross entropy loss w/ clean | 68.57 | 78.38 | 71.99 | 75.50 | 68.57 | 78.38 |
| Cross entropy loss w/ noisy | 56.88 | 64.13 | 58.59 | 62.75 | - | - |
| No pairwise link | 63.67 | 73.19 | 66.18 | 71.78 | 58.01 | **67.84** |
| CRF w/o hidden | 64.26 | 73.23 | 67.73 | 71.78 | 59.04 | 67.22 |
| CRF w/ hidden | 65.73 | 74.04 | 68.35 | 71.92 | 59.19 | 67.33 |
| CRF w/o $x - y$ link | **66.61** | **75.00** | **69.89** | **73.16** | **60.97** | 67.57 |
| CRF w/o $x - y$ link ($\alpha = 0$) | 48.53 | 56.53 | 56.76 | 56.39 | 47.25 | 58.74 |
| Misra et al. [4] | - | - | - | 66.8 | - | - |
| Fang et al. [36] reported in [4] | - | - | - | 63.7 | - | - |

$q$ differs from $p_{aux}$. Since, we have access to the clean labels in the COCO dataset, we examine the accuracy of $q$ in terms of predicting clean labels on the noisy training set ($D_N$) using the mAP measurement at the beginning and end of training the CRF-CNN model (ResNet-50 on large images). We observed that at the beginning of training, when $\alpha$ is big, $q$ is almost equal to $p_{aux}$, which obtains 49.4% mAP on this set. As training iterations proceed, the accuracy of $q$ increases to 69.4% mAP. Note that the 20.0% gain in terms of mAP is very significant, and it demonstrates that combining the auxiliary distribution with our proposed CRF can yield a significant performance gain in inferring latent clean labels. In other words, our proposed model is capable of cleaning the noisy labels and proposing more accurate labels on the noisy set as training continues. Please refer to our supplementary material for a qualitative comparison between $q$ and $p_{aux}$.

### 4.3 CIFAR-10 Dataset

We also apply our proposed learning framework to the object classification problem in the CIFAR-10 dataset. This dataset contains images of 10 objects resized to 32x32-pixel images. We follow the settings in [9] and we inject synthesized noise to the original labels in training. Moreover, we implement the *forward* and *backward* losses proposed in [9] and we use them to train ResNet [33] of depth 32 with the ground-truth noise transition matrix.

Here, we only train the variant of our model shown in Fig. 2.c that can be trained analytically. For the auxiliary distribution, we trained a simple linear multinomial logistic regression representing the conditional $p_{aux}(\hat{y}|y)$ with no hidden variables ($h$) . We trained this distribution such that the output probabilities match the ground-truth noise transition matrix. We trained all models for 200 epochs. For our model, we anneal $\alpha$ from 8 to 1 in 10 epochs. Similar to the previous section, we empirically observed that it is better to stop annealing $\alpha$ before it reaches zero. Here, to compare our method with the previous work, we do not work in a semi-supervised setting, and we assume that we have access only to the noisy training dataset.

Our goal for this experiment is to demonstrate that a simple variant of our model can be used for training from images with only noisy labels and to show that our model can clean the noisy labels. To do so, we report not only the average accuracy on the clean test dataset, but also the *recovery accuracy*. The recovery accuracy for our method is defined as the accuracy of $q$ in predicting the clean labels in the noisy training set at the end of learning. For the baselines, we measure the accuracy of the trained neural network $p(\hat{y}|x)$ on the same set. The results are reported in Table 2. Overall, our method achieves slightly better prediction accuracy on the CIFAR-10 dataset than the baselines. And, in terms of recovering clean labels on the noisy training set, our model significantly outperforms the baselines. Examples of the recovered clean labels are visualized for the CIFAR-10 experiment in the supplementary material.

## 5   Conclusion

We have proposed a general undirected graphical model for modeling label noise in training deep neural networks. We formulated the problem as a semi-supervised learning problem, and we proposed a novel objective function equipped with a regularization term that helps our variational distribution

Table 2: Prediction and recovery accuracy of different baselines on the CIFAR-10 dataset.

| Noise (%) | Prediction Accuracy (%) | | | | | Recovery Accuracy (%) | | | | |
|---|---|---|---|---|---|---|---|---|---|---|
| | 10 | 20 | 30 | 40 | 50 | 10 | 20 | 30 | 40 | 50 |
| Cross entropy loss | 91.2 | 90.0 | 89.1 | 87.1 | 80.2 | 94.1 | 92.4 | 89.6 | 85.2 | 74.6 |
| Backward [9] | 87.4 | 87.4 | 84.6 | 76.5 | 45.6 | 88.0 | 87.4 | 84.0 | 75.3 | 44.0 |
| Forward [9] | 90.9 | 90.3 | 89.4 | 88.4 | 80.0 | 94.6 | 93.6 | 92.3 | 91.1 | 83.1 |
| Our model | **91.6** | **91.0** | **90.6** | **89.4** | **84.3** | **97.7** | **96.4** | **95.1** | **93.5** | **88.1** |

infer latent clean labels more accurately using auxiliary sources of information. Our model not only predicts clean labels on unseen instances more accurately, but also recovers clean labels on noisy training sets with a higher precision. We believe the ability to clean noisy annotations is a very valuable property of our framework that will be useful in many application domains.

**Acknowledgments**

The author thanks Jason Rolfe, William Macready, Zhengbing Bian, and Fabian Chudak for their helpful discussions and comments. This work would not be possible without the excellent technical support provided by Mani Ranjbar and Oren Shklarsky.

## Footnotes

[1]Each sample is assumed to belong to only one class.

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
