[Supplementary Material]

# Supplementary Material:
# Toward Robustness against Label Noise in Training Deep Discriminative Neural Networks

**Arash Vahdat**
D-Wave Systems Inc.
Burnaby, BC, Canada
avahdat@dwavesys.com

## 1  Visualization

As shown in the E step in Sec. 3.3, the variational distribution $q$ infers latent clean labels by combining information from both the image-based CRF-CNN model $p_{\boldsymbol{\theta}}(\hat{\boldsymbol{y}}, \boldsymbol{h} | \boldsymbol{y}, \boldsymbol{x})$ and the label-based auxiliary distribution $p_{aux}^{\alpha}(\hat{\boldsymbol{y}}, \boldsymbol{h} | \boldsymbol{y})$. In our experiments, we observe that in general $q$ proposes clean labels more accurately than the auxiliary distribution. Fig. 1 compares $q$ against $p_{aux}$ in terms of its ability to infer clean labels for a few instances in the noisy training set ($D_N$) for the COCO experiment with actual Flickr tags. In Fig. 2, examples of the recovered clean labels are visualized for the CIFAR-10 experiment.

| | |
|---|---|
| **Flickr** | ∅ |
| $p_{aux}$ | person |
| $q$ | skateboard, person |
| **clean** | skateboard, person |

| | |
|---|---|
| **Flickr** | ∅ |
| $p_{aux}$ | person |
| $q$ | person, baseball glove, baseball bat |
| **clean** | person, baseball glove, baseball bat, sports ball, chair, bench |

| | |
|---|---|
| **Flickr** | 2009, miami |
| $p_{aux}$ | person |
| $q$ | person, tennis racket |
| **clean** | person, tennis racket |

| | |
|---|---|
| **Flickr** | uploaded:by=flickr_mobile, flickriosapp:filter=NoFilter |
| $p_{aux}$ | person |
| $q$ | person, surfboard |
| **clean** | person, surfboard |

| | |
|---|---|
| **Flickr** | computer |
| $p_{aux}$ | ∅ |
| $q$ | laptop, mouse, tv, keyboard |
| **clean** | laptop, mouse, tv, keyboard |

| | |
|---|---|
| **Flickr** | square, iphoneography, square format, instagram app, uploaded:by=instagram |
| $p_{aux}$ | ∅ |
| $q$ | cup, dining table, bottle, bowl |
| **clean** | cup, dining table, bottle, bowl, spoon, hot dog |

| | |
|---|---|
| **Flickr** | food, square, square format, nikon, white, orange, fruit, color, India, photography, table, project365, bowl, colour, wood, 50mm, nikkor, bokeh |
| $p_{aux}$ | orange, apple, banana |
| $q$ | orange, bowl, dining table |
| **clean** | orange, bowl |

| | |
|---|---|
| **Flickr** | home, light, photo, art, chair, room, table, architecture, apartment, interior, couch, decor, beauty, design, live, lamp, indoor, furniture, relaxed, sofa, flooring, modern |
| $p_{aux}$ | chair, couch, vase, book, dining table, sink, clock, bed, potted plant |
| $q$ | chair, couch, vase, book |
| **clean** | chair, dining table, tv |

Figure 1: Visualization of inferred labels for a few instances in the noisy training set ($D_N$) of the COCO dataset. Flickr labels represent the noisy labels extracted from Flickr tags, whereas clean labels are the true labels ignored during training. $p_{aux}$ and $q$ correspond to the labels that are extracted using these distribution by thresholding at 0.5. The auxiliary distribution $p_{aux}$ tends to assign the label "person" to the images with no tag while $q$ adds more clean labels. In the last two images, $q$ removes a few unrelated labels.

(a) cat → dog

(b) dog → cat

(c) automobile → truck

(d) horse → deer

Figure 2: Our proposed model can recover clean labels in the noisy training dataset. Here, corrupted instances are visualized for different categories in the CIFAR-10 training dataset. Sub-figures (a) through (d), captioned with *annotated label → inferred label*, represents the instances that are labeled with the annotated label but have been assigned to the inferred label by our proposed variational distribution $q$. In this visualization, images are sorted based on the confidence of $q$ for the inferred label from left to right and top to bottom, and the mistaken instances are marked with the red frame. The probability that $q$ assigns for the inferred label is typically very high ($> 0.9$) for these images, which indicates that $q$ is confident in changing the noisy labels.