[Reviews · NeurIPS 2017]

Reviewer 1



This submission employs a conditional random field model to handle the label noise. EM algorithm is designed and empirical experiments verified the effectiveness. However, in the current version, we don't theoretically know why modern deep convolutional neural networks can gain robustness against label noise. Pls provide some theoretical guarantees. Also, the submission contains too many tricks, e.g., conditional random filed, a small set of clean labels, semi-supervised learning, and auxiliary distributions. We don't know which one really helps, which significantly weaken the paper. I also have a minor concern about how to choose the size of the subset of clean labels.

Reviewer 2



The paper proposes a method for incorporating data with noisy labels into a discriminative learning framework (assuming some cleanly labeled data is also available), by formulating a CRF to relate the clean and noisy labels. The CRF sits on top of a neural network (e.g., CNN in their experiments), and all parameters (CRF and NN) are trained jointly via an EM approach (utilizing persistent contrastive divergence). Experiments are provided indicating the model outperforms prior methods on leveraging noisy labels to perform well on classification tasks, as well as inferring clean labels for noisily-labeled datapoints. The technique is generally elegant: the graphical model is intuitive, capturing relationships between the data, clean/noisy labels, and latent "hidden" factors that can encode structure in the noisy labels (e.g., presence of a common attribute described by many of the noisy labels), while permitting an analytically tractable factorized form. The regularization approach (pre-training a simpler unconditional generative model on the cleanly labeled data to constrain the variational approximation during the early part of training) is also reasonable and appears effective. The main downside of the technique is the use of persistent contrastive divergence during training, which (relative to pure SGD style approaches) adds overhead and significant complexity. In particular the need to maintain per-datapoint Markov chains could require a rather involved setup for truly large scale noisy datasets with hundreds of millions to billions of datapoints (such as might be obtained from web-based or user-interaction data). The experiments are carefully constructed and informative. I was particularly happy to see the ablation study estimating the contribution of different aspects of the model; the finding that dropping the data <-> noisy-label link improves performance makes sense in retrospect, although wasn't something I was expecting. I would have been curious to see an additional experiment using the full COCO dataset as the "clean" data, but leveraging additional Flickr data to improve over the baseline (what would be the gain?). The paper was quite readable, but could benefit from another pass to improve language clarity in a few places. For instance, the sentence beginning on line 30 took me a while to understand. Overall, I think the paper is a valuable contribution addressing an important topic, so I recommend acceptance.

Reviewer 3



The authors propose a strategy for dealing with label noise, particularly in the setting where there are multiple target labels per instance, which distinguishes it from much of the previous work in the area. Overall, the paper is well-written and clear to follow. I only had a few minor comments which are listed below: - In Equation 3: I think it would be interesting to examine whether weighting the relative contributions from the clean and noisy labeled data affects overall performance. I think this would be particularly interesting if the relative amounts of clean vs. noisy labeled data are varied. - In the results section, could the authors also report the results in terms of p-values after performing a significance test so that it's clearer how strong the relative differences between the various systems are. In particular, at Line 274 the authors mention, "Note that the 20.0% gain in terms of mAP is very significant". Please mention p-values and how statistical significance was computed here. - There have been a few previous works that have examined incorporating neural networks to predict CRF potentials from x, which the authors could consider citing as part of the relevant previous work, e.g.: * J. Peng, B. Liefeng, J. Xu. "Conditional neural fields." Advances in neural information processing systems. 2009. * T.-M.-T. Do and T. Artieres. "Neural conditional random fields." International Conference on Artificial Intelligence and Statistics (AI-STATS). 2010. * R. Prabhavalkar and E. Fosler-Lussier. "Backpropagation training for multilayer conditional random field based phone recognition." IEEE International Conference on Acoustics, Speech, and Signal Processing (ICASSP). 2010. * L. Van der Maaten and M. Welling and L. K. Saul. "Hidden-unit conditional random fields." International Conference on Artificial Intelligence and Statistics. 2011. Minor comments and Typographical errors: 1. Line 20: "Most previous work in this area have focused" --> "Most previous work in this area has focused". Also, please cite relevant works (which appear later in the paper here as well) 2. Line 59-60: "Misra et al. [4] targets the image multilabeling problem but models" --> "Misra et al. [4] target the image multilabeling problem but model" 3.Line 64: "obtained from web" --> "obtained from the web" 4. Line 270: "mAP". Please introduce terms before using acronyms for the first time.